# Combining Sparse and Dense Features to Improve Multi-Modal Registration for Brain DTI Images

**DOI:** 10.3390/e22111299

**Published:** 2020-11-14

**Authors:** Simona Moldovanu, Lenuta Pană Toporaș, Anjan Biswas, Luminita Moraru

**Affiliations:** 1Department of Computer Science and Information Technology, Faculty of Automation, Computers, Electrical Engineering and Electronics, Dunarea de Jos University of Galati, Galati 47 Domneasca Str., 800008 Galati, Romania; simona.moldovanu@ugal.ro; 2The Modelling & Simulation Laboratory, Dunarea de Jos University of Galati, Galati 47 Domneasca Str., 800008 Galati, Romania; Lenuta.Pana@ugal.ro; 3Department of Chemistry, Physics & Environment, Faculty of Sciences and Environment, Dunarea de Jos University of Galati, 47 Domneasca Str., 800008 Galati, Romania; 4Department of Physics, Chemistry and Mathematics, Alabama A&M University, Normal, AL 35762-4900, USA; biswas.anjan@gmail.com; 5Department of Mathematics, King Abdulaziz University, Jeddah 21589, Saudi Arabia; 6Department of Applied Mathematics, National Research Nuclear University, 31 Kashirskoe Hwy, 115409 Moscow, Russia

**Keywords:** image registration, mutual information, histogram-oriented gradients, sparse and dense features, fiducial registration error

## Abstract

A new solution to overcome the constraints of multimodality medical intra-subject image registration is proposed, using the mutual information (MI) of image histogram-oriented gradients as a new matching criterion. We present a rigid, multi-modal image registration algorithm based on linear transformation and oriented gradients for the alignment of T2-weighted (T2w) images (as a fixed reference) and diffusion tensor imaging (DTI) (*b*-values of 500 and 1250 s/mm^2^) as floating images of three patients to compensate for the motion during the acquisition process. Diffusion MRI is very sensitive to motion, especially when the intensity and duration of the gradient pulses (characterized by the *b*-value) increases. The proposed method relies on the whole brain surface and addresses the variability of anatomical features into an image stack. The sparse features refer to corners detected using the Harris corner detector operator, while dense features use all image pixels through the image histogram of oriented gradients (HOG) as a measure of the degree of statistical dependence between a pair of registered images. HOG as a dense feature is focused on the structure and extracts the oriented gradient image in the x and y directions. MI is used as an objective function for the optimization process. The entropy functions and joint entropy function are determined using the HOGs data. To determine the best image transformation, the fiducial registration error (FRE) measure is used. We compare the results against the MI-based intensities results computed using a statistical intensity relationship between corresponding pixels in source and target images. Our approach, which is devoted to the whole brain, shows improved registration accuracy, robustness, and computational cost compared with the registration algorithms, which use anatomical features or regions of interest areas with specific neuroanatomy. Despite the supplementary HOG computation task, the computation time is comparable for MI-based intensities and MI-based HOG methods.

## 1. Introduction

Brain image registration is a fundamental component of the medical image analysis pipeline and consists of an optimization of the spatial correspondence between pairs of images by aligning corresponding structures to ease the comparison between different brain images. During the acquiring of stacks of 2D slices in real-time MRI, it is desirable to reduce the scan time and to avoid the slice crosstalk artifacts. The images of interest are acquired either at different times (multi-temporal analysis) or using various devices/sensors (multi-modal analysis) and may belong to the same patient or to different patients [1,2,3,4,5,6,7,8,9,10]. Moreover, the same structure taken from different viewpoints (multi-view analysis) may be considered.

However, despite all these efforts, the diversity of images to be registered and the wide range of types of image degradations cause a universal registration method to be intractable.

Being an automatic diagnosis tool, image registration adds value to imagery information media by allowing MR image modalities like T2w and diffusion tensor imaging (DTI) brain images to be analyzed in the same coordinate system, especially to correct the motion [6,7,8]. Grigorescu et al. [7] proposed a deep learning registration approach for the registration of T2w and DTI images belonging to infants born and scanned at different gestational ages. The proposed T2w + DTI registration model provided better results in terms of aligning subcortical structures. The performance of the proposed registration model was compared to baseline model trained T2w data and the differences between the moved and fixed fractional anisotropy maps. The benefits of using brain image multi-modal registration consist of aligning corresponding structures to ease the comparison between different brain images. Once the images are aligned, they share the same coordinate system and the difference between test and reference images can be easily analyzed. Diffusion MRI yields information on the underlying organization of the brain white matter and the connections inside the brain. T2w images are useful for detecting edema, revealing white matter lesions, and provide structural information.

Such a T2w-DTI multi-modal registration approach is the main interest of this paper, as the proposed method uses the whole brain surface and addresses the variability of anatomical features into image stack. We present a rigid and multi-modal image registration algorithm based on linear transformation and oriented gradients for the alignment of T2w images and DTI (*b*-values of 500 and 1250 s/mm^2^) images.

Several recent papers have addressed the importance of developing image registration techniques and made it beneficial in medical diagnosis or treatment using a rigid and linear registration algorithm applied to brain images [10,11,12]. Roura et al. [10] studied a co-registration method between structural T1-weighted (T1w) scans and fractional anisotropy maps of DTI images. A multi-channel registration algorithm was implemented and tested using 100 simulated brain atrophy images. The experimental results indicated that the multi-channel registration approach and fractional anisotropy maps provide significant improvements in alignment accuracy over single-channel or T1w-based pipelines. Lin et al. [11] reported a sound performance comparison of the commercially available MRI analysis/image registration software packages for information of clinical users and for further development of improved algorithms for clinical use. They used 20 patients and four sequences: T2w, FLAIR (fluid attenuated inversion recovery), susceptibility-weighted angiography, and T1 postcontrast. Zhang et al. [12] proposed a knowledge-based approach for registration accuracy improvement using so-called “mediator images”, which act as intermediates between a reference and registered images. Linear transformations were performed between reference and mediator and mediator and registered images and a library of intermediate images (mediators) was proposed. To overcome the drawback of increased computational time, the authors reduced the size of the library by clustering. They claim a clear improvement in registration accuracy.

The utility of image registration in medical diagnosis or treatment is often influenced by the modality of acquisition. Nowadays, the multi-modal registration approach enhances the reliability of the correspondence and comparison across subjects [13,14]. Guyader et al. [13] proposed so-called groupwise registrations, which are able to register two or more images without the need of a reference image. The total correlation replaces the mutual information as an objective function for optimization. Registration experiments were conducted on a dynamic CT (computed tomography) imaging dataset, and on five quantitative magnetic resonance imaging datasets. In the framework of groupwise image registration, the proposed total correlation measure provides registration results comparable to those of other previously reported experiments. Goubran et al. [14] combined histologically cleared volumes with connectivity atlases and MRI to investigate the global structural and network changes following an ischemic stroke. They implemented this feature for multimodal interrogation of brain connectivity by registered imaging of cleared volumes with MR/CT and the Allen atlas. The ultimate goal was to demonstrate tract-level histological changes of stroke by instrumentality of multi-modal registration.

Image similarity metrics such as mean squared difference (MSD) or normalized cross correlation (CC) are largely used as optimization functions when image pairs with similar contrast/unimodal registration are used for registration [15,16,17]. The unimodal image registration just uses the assumption that corresponding pixels have similar intensity values. On the contrary, in multimodal registration problems, the same structure may display different intensities. For images acquired with different modalities, metrics such as mutual information (MI) or correlation ration (CR) are indicated because the same brain structures may have quite different gray values in the multi-modal case.

To estimate the best parameters of rigid transformation for an optimal registration and to solve an ill-posed problem due to sparse information on intra- and inter-correspondence of scenes in multiple images, the mutual information (MI) criterium, which takes intensity information into account, is a valuable solution [18,19,20]. MI is a measure of common information in source and target images. Fan et al. [19] highlighted that the intensity discretization procedure, before the accumulation of statistical entropy used in MI computation, leads to poor performance in registration. To overcome this issue, they embedded a function of the gradient information and prioritized strong gradient regions over the small gradient regions. The results have indicated that the image registration becomes stable. Yang et al. [20] proposed a new method that combines the normalized mutual information with spatial information for nonrigid medical image registration. The algorithm was validated on a simulated brain image with single-modality and multi-modality. When both the spatial and intensity differences of different tissues with different imaging modes are considered, satisfactory registration results are obtained. However, the performance of MI is affected when the local intensity variations manifest.

The accuracy of the registration and the quantification of the registration error is performed usually by the so-called fiducial registration error (FRE). It analyzes the relationship between the position and orientation parameters of rigid transformations [21,22,23,24]. FRE is a sum-of-squared difference for pairs of selected points of interest within the source image space and the corresponding points of interest within the target image; these points are the fiducials and are used to assess the registration validation.

We are unaware of any previous studies that have registered images across T2w and DTI (*b*-values of 500 and 1250 s/mm^2^, i.e., different gradient amplitude, duration, and time interval between gradient pulses) modalities. We report an intra-patient registration method of T2w and DTI data across the whole brain images of three datasets, one for a healthy patient, one for a patient with right parietal lobe hemorrhage, and one for a patient with ischemic stroke. Each dataset includes three T2w images/15 slices per image, three DTI (500 s/mm^2^) images/15 slices per image, and three DTI (1250 s/mm^2^) images/15 slices per image. The multiple registration algorithm is applied using each T2w stack sequence as fixed reference images and stacks of DTI images for *b*-values of 500 and 1250 s/mm^2^ are used for registration. The relative starting and relative ending positions of the image stacks ensure the premises of a diversity of spatial and intensity information, leading to a robust registration. Hence, we investigate how to choose optimal transformations through a robust and user-independent decision metric to improve the accuracy of performance in registration algorithms. The contributions of this paper are as follows:-Sparse/geometrical and dense features are considered. The spatial information is extracted as “sparse features” by the instrumentality of the Harris corner features. They are robust to global changes of intensity and provide an initial coarse registration. Further, the HOG feature descriptor takes into consideration the gradient orientation in localized portions of an image and organizes the gradients into histograms. The magnitude of gradients is large around the edges and corners. The histogram of gradients (i.e., the magnitude and gradient directions, at every pixel) provides the so-called “dense features”. Dense features search for the degree of statistical dependence between the intensities in a pair of registered images;-To reduce the uncertainty of the transformation, multi-source information that uses image information such as edge features or their gradients helps to quantify the similarity between images. The key contribution of this paper is the method by which the entropy and MI are estimated. Thus, histograms of oriented gradients (HOGs) are used to compute entropy of source/target images, as well as the joint entropy of both images. First, the optimal translation parameter *τ* was identified using the smallest Euclidian distance (ED) values; this step ensures a coarse registration to eliminate significant scale differences. Then, the rotational parameters of the rigid transformation (Θ) followed by HOG feature extraction are iteratively generated by the MI in order to find the best matches between reference registered images. It provides the final, fine registration.-The accuracy of the proposed registration method is checked by using a fiducial registration error (FRE) driven by the minimization of the distance between the selected fiducials/Harris corners in both images after registration.

We show that a T2w image together with features from a DTI image can be used to properly align subjects with improved accuracy. Moreover, the proposed method can be implemented in the problem of the automated brain DTI image follow up, i.e., detection of abnormalities by comparing different images of the same patient using an older brain DTI image as a reference and highlighting the possible presence of abnormalities at the time of evolution.

The paper is organized as follows. Section 2 outlines the proposed methods to brain images’ registration. Section 3 describes and discusses the results, and Section 4 summarizes the conclusion of the present work.

## 2. Materials and Methods

### 2.1. Transformation Model. Harris Corner Detection. Rigid Registration Using a Global Mapping Method and Histogram of Oriented Gradients

The image registration method is formulated as a parametric optimization problem to maximize the similarity between a source image *I_s_* and target image *I_t_* through a transformation T:(1)$$T=\arg\underset{\Theta ,\;\tau }{\max}Metric\left[I_{s},\;I_{t}\circ \;TR_{global}\right]$$
where $\;TR_{global}$ denotes the transformation model, which is adopted here as a global rigid transformation parametrized by translation *τ* and rotation Θ. *Metric* is a similarity measure identified for this study with mutual information (MI). A global mapping method is linear transformation-based and allows the calculation of translational and rotational vectors when the integrity and consistency of the brain structures are maintained. In this case, the registration is based on a given set of known point pairs while the brain volume, size, and shape are preserved. This kind of transformation belongs to a low-dimensional parametric class of deformations and mainly corrects the head movement in various image stacks belonging to the same patient.

$\;TR_{global}$ maps all the points from the space of a source image to the space of a target image:(2)$$TR_{global}\left(x\right)=\Theta \left(x\right)+\tau \left(x\right)$$
where $x=\left(\begin{array}{c} x\\ y \end{array}\right)$ denotes the coordinates of *i*th pixel in the image, $\Theta =\left(\begin{array}{cc} \cos\theta & -\sin\theta \\ \sin\theta & \cos\theta \end{array}\right)$ accounts for the rotation matrix by angle *θ*, and $\tau =\left(\begin{array}{c} \tau_{x}\\ \tau_{y} \end{array}\right)$ is the translation vector over the *x* and *y* axis.

In order to perform the translation operation, the interest points, such as corners, are determined using the Harris corner detector operator (*R*) [25,26]. Corners are the key features of the morphology of brain images. The Harris operator is computed from the local autocorrelation function of the partial derivative of intensity in the *x* and *y* directions and relied on aggregated gradients. They represent the sparse features.

Into the image, corners are regions with large variation in intensity in all directions, and represent the more stable features across scaling/translation, rotation, or illumination, for example. The Harris corner detection is based on finding the difference in intensity for the shift (*u*, *v*) in all directions:(3)$$S\left(u,\;v\right)=\underset{x}{\sum }\underset{y}{\sum }w\left(x,\;y\right){\left(I\left(x,\;y\right)-I\left(x-u,\;y-v\right)\right)}^{2}$$
where I(x, y) is the intensity and I(x−u, y−v) denotes the shifted intensity. The window function w(x, y) is either a rectangular or Gaussian window. For corner detection, the function S(u, v) must be maximized using the Taylor expansion and some math operations. A 2 × 2 matrix for image derivatives in the *x* and *y* directions allows one to compute the eigenvalues $\lambda_{1}$ and $\lambda_{2}$, respectively. In this way, we obtain the directions for both the largest and smallest variation of S(u, v). The corner response is as follows:(4)$$R=\;\lambda_{1}\lambda_{2}-k\left(\lambda_{1}+\lambda_{2}\right)$$
where *k* is set empirically between 0.04 and 0.06. The region is a corner when *R* is large; this means that $\lambda_{1}$ and $\lambda_{2}$ are large and $\lambda_{1}\approx \lambda_{2}$. The strong corners are selected in the source and registered images.

These corner features are used to put an image pair into correspondence. Initial, a coarse registration is performed through a translation of −5 to +5 pixels with a step size of 0.5 pixels along the *x* and *y* axis. Then, the Euclidian distance (ED) between similarly strong corners in each pair image is computed. The new position of the translated registered image is evaluated according to ED values. The best new translated position is determined using the smallest ED values.

After it is established that two images have been correctly matched by a certain value of translation *τ*, a rotation is performed and histograms of gradients are generated using the HOG algorithm. First, the image is divided into 8 × 8 patches. Then, the gradients for every pixel in the x and y directions are calculated. Moreover, the magnitude and direction for each pixel value are determined. To minimize the influence of illumination effects, the gradients were normalized by considering 16 × 16 blocks. Thus, four 8 x 8 patches are combined to create a 16 × 16 block to extract HOG features. Then, histograms are generated using gradients and orientation for each block.

### 2.2. Mutual Information

The probability distribution functions of gradients are used to compute the entropy of source/target images, as well as the joint entropy of both images, for each rotation parameter [27,28,29]. They represent the dense features. This step is the fine registration part. MI came from the information theory and assumes a statistical relationship between two discrete random variables, which, in our case, are image intensities *p* and *q* [30,31]:(5)$$MI=H\left(p\right)+H\left(q\right)-H\left(p,q\right)\;H\left(p\right)=-\sum_{i}p_{i}\log p_{i},\;H\left(q\right)=-\sum_{j}q_{j}\log q_{j},\;H\left(p,\;q\right)=-\sum_{i,j}p_{ij}\log p_{ij}$$
where *H* denotes the Shannon entropy, $p_{i}$ and $q_{i}$ are the marginal probability distribution of HOG intensity occurring in the source (registered) image, and $p_{ij}$ is the joint probability distribution of both occurring at the same place. Moreover, we assume a whole brain similarity of two images. The gradient-related entropy makes the MI similarity measure more definitive for each pixel and for its intensity level, helping to avoid local minima. In this way, the robustness to illumination change or small distortions is assured.

MI analyzes the feature vectors generated by the HOG algorithm for all locations in the image and provides certain informational correlation between images. The analysis was done for an optimal translation parameter and for a rotation in the range *θ*∈[−5°, +5°], with the step of variation being 0.5°. MI is larger when the analyzed images are more similar and are correctly aligned.

### 2.3. Assessment of Registration Accuracy

The fiducial registration error (FRE) metric, which is a sum-of-squared difference between the position of fiducial/corner points in the source image (R) viewed as the ground truth and the position of fiducial/corner points in the registered image ($\tilde{R}$) obtained with the registration method, was used [22,23,24]:(6)$$FRE=\frac{1}{\left|\Omega \right|}\overset{\left|\Omega \right|}{\underset{k=1}{\sum }}{\left(R_{k}-{\tilde{R}}_{k}\right)}^{2}$$

The selected fiducial points (|Ω|) are spread evenly over the whole brain image and represent the Harris corners. We used |Ω| = 30 corner points for the T2w and DTI (*b*-values of 500 and 1250 s/mm^2^) images, respectively. |Ω| represents the minimum number of corresponding corner points identified in all the analyzed images and they have low mobility and high importance, according to the ED values. A low FRE value signifies good registration accuracy. The accuracy of the registration method was assessed as the root mean square (RMS), mean, and standard deviation of the FRE values after the registration.

### 2.4. Dataset Acquisition

All datasets were recorded by the authors. MRI scans were performed using a 1.5 T MRI scanner (Philips Medical Systems, Best, Netherlands) in the Radiology Department of “Sfântul Andrei” Hospital, Galati (Romania). DTI sequences were acquired using a system with six-channel sensitivity encoding (SENSE) for faster scanning (FS = 1.5). The imaging parameters were as follows: 3D gradient echo with echo time (TE) ranging from 83 to 110 ms; repetition time (TR) ranging from 6500 to 7800 ms; bandwidth = 1070 Hz/pixel; flip angles (2- and 6-); voxel resolution ranging from 2.5 to 3.0 mm; and slice thickness = 4 mm. The acquisition matrix was 128 × 128.

Three datasets—S1 for a healthy patient (male, 36 y), S2 for a patient (male, 68 y) with right parietal lobe hemorrhage and a small sequela in the right hemisphere, and S3 for a patient with ischemic stroke (male, 74 y)—were investigated. Each dataset includes three T2w images/15 slices per image, three DTI (*b*-values = 500 s/mm^2^) images/15 slices per image, and three DTI (*b*-values = 1250 s/mm^2^) images/15 slices per image. The procedure was applied to pairs of T2w-DTI (*b*-values = 500 s/mm^2^) and T2w-DTI (*b*-values = 1250 s/mm^2^). A T2w scan was performed without coil using identical parameters. The images were skull stripped and no other preprocessing was applied.

Approval for the study was obtained from the Research Ethics Committee of the “Dunărea de Jos” University of Galați and “Sfântul Andrei” Hospital. Voluntary and informed consent was obtained in writing from each patient involved.

## 3. Results and Discussion

The proposed model is an end-to-end model without using any preprocessing methods or denoising techniques. Each target image (DTI, *b*-values = 500 and 1250 s/mm^2^) was aligned to the source image (T2w) directly using the proposed method. Figure 1 shows the flow chart of this new proposed method. For the sparse feature determination, an experiment based on Harris corner detector operator was carried out. The Harris operator was used to extract corner and edge information by setting k = 0.05 and T = 1500. These parameters were set empirically to ensure the best registration performance. Two translations in horizontal and vertical directions, respectively, were performed. The strong corners are robust to global changes of intensity and are selected in the source (*R*) and registered images ($\tilde{R}$), and ED between similar strong corners in each pair of images is computed when the registered image is translated on the *x* and *y* axis with an imposed step. The optimal translation parameter τ ensures a coarse registration and eliminates significant scale differences. Then, the fine registration based on the rotation is performed in the range *θ*∈[−5°, +5°] with a step of variation of 0.5°, and histograms of gradients (HOGs, HOGt) for the source (s) and target (t) images are determined. The fine registration stage amends and improves the misalignment and refines the registration performance through a local approach by HOG. Then, MI and FRE are computed.

The experiment results for the translation of the registered images are provided in Figure 2 and Figure 3. In the coarse registration step, the average number of sparse features/corners per image varied as follows: for S1, from 40 to 181 for T2w-DTI (*b*-values = 500 s/mm^2^) and from 40 to 166 for T2w-DTI (*b*-values = 1250 s/mm^2^), respectively. For S2, from 38 to 167 for T2w-DTI (*b*-values = 500 s/mm^2^) and from 35 to 152 for T2w-DTI (*b*-values = 1250 s/mm^2^), respectively. For S3, from 42 to 160 for T2w-DTI (*b*-values = 500 s/mm^2^) and from 39 to 144 for T2w-DTI (*b*-values = 1250 s/mm^2^), respectively. They are spread evenly over the brain surface and are located in that spots where a large variation in intensity in all the directions manifests. These selected corners have similar locations and belong to the same regions in the pair images. The locations of these corners have certain variability, but the mobility of corners is limited by the brain structure to a narrow range. In our case, the low mobility of corners indicates high importance values of corners and helps generate matched Harris corner pairs. The optimal translation parameters $\tau =\left(\begin{array}{c} \tau_{x}\\ \tau_{y} \end{array}\right)=\left(\begin{array}{c} -4.5\\ -4.5 \end{array}\right)$ for S1, $\tau =\left(\begin{array}{c} \tau_{x}\\ \tau_{y} \end{array}\right)=\left(\begin{array}{c} -2.5\\ -2.5 \end{array}\right)$ for S2, and $\tau =\left(\begin{array}{c} \tau_{x}\\ \tau_{y} \end{array}\right)=\left(\begin{array}{c} 2.5\\ 2.5 \end{array}\right)$ for S3, respectively, were identified using the smallest ED values between similar strong corners. These parameters ensure a coarse registration to eliminate significant scale differences. This coarse registration step shortens the exploration range in the transformation model.

For each completed rigid transformation image, HOG, as a fine-scale gradient computation technique, is applied and the gradients and orientations of the edges, for all locations, are collected to form the feature vectors. These features generate the HOG histogram and provide certain informational correlation between images. Figure 4 shows examples of HOG features as the final bins and their magnitude. The image intensities and change in intensities are exceptionally described for each local cell. A block size of 8 × 8 contains abundant shape information and the number of HOG features is 34,596.

Figure 5 shows examples of the MI-based HOG comparable graphs. The maxima for the image-wise MI scores is obtained in terms of rotation and translation parameters, as follows: for S1 (−2°, −4.5 mm, −4.5 mm), for S2 (2°, −2 mm, −2 mm), and for S3 (2.5°, 2.5 mm, 2.5 mm), respectively. It is well known that MI would not yield to a precise alignment of the two datasets mainly because of the joint entropy, which is the main hindrance in the registration, by indicating how much information one signal has about another. However, the individual input’s entropy corrects this issue and recommends MI as the best similarity measure for image registration. MI computed based HOG features consider the spatial information about the pixel, thus the main limitation of MI is addressed and overcome.

The FREs between corrected displacements (translation + rotation) and initial positions were used as a metric to measure the accuracy of the registration. Image registration is successful for smaller values of FRE (Figure 6). FRE was 2.31 ± 1.90 pixel (or 0.6111 ± 0.5027 mm) for S1; 2.1 ± 1.65 pixel (or 0.5556 ± 0.4365 mm) for S2, and 2.48 ± 1.56 pixel (or 0.6561 ± 0.4127 mm) for S3, respectively. This is the smallest registration error. Comparing the MI and FRE result, the T2w-DTI registration has angles of rotation of −2°, +2°, and +2.5° as optimal parameters. The FRE results show that the proposed MI-based HOG method provided accurate results and the corresponding points of the different images have the same spatial and anatomic positions. Hoelper et al. [32] reported a whole-brain volume registration error ranging from 0.7 to 2 mm, depending in the region of the brain when 25 anatomic landmarks were placed in T1w and T2w brain volumes. Grigorescu et al. [7] reported the best average Dice scores for their cross-validation study in the T2w-DTI case for all studied subcortical structures in comparison with the T2w-only model (0.88 against 0.84). Moreover, when the same model is applied to the warped and fixed fractional anisotropy (FA) maps better registration results were obtained for the T2w-DTI case.

Although MI computed as a statistical intensity relationship between corresponding pixels in both the source and target image is largely accepted as a gold standard similarity measure for multimodal image registration, there are no gold standard registrations available for T2w-DTI images registration-based HOG, so the results of the proposed method are compared to the MI based on image intensities of corresponding pixels. Figure 7 shows examples of the MI based on image intensities of corresponding pixels graphs.

This MI-based intensity similarity measure not only considers the intensity statistical characteristics of the global consistency of images, but also the spatial information is ignored. As a natural consequence, it cannot notice the complex interactions among the pixel intensities. Some small local maxima exist. They are mainly caused by a local match between pixel intensities driven by using different imaging principles or by the large intensity difference of different tissues in different imaging modes. Usually, these local extremes lead to misregistration. MI-based HOGs reduce the risk of falling into the local maxima and improve the robustness of the registration compared with the traditional MI measure.

It can be concluded that the MI measure does not have a good registration result when it is calculated using intensity information from pixel to pixel alone. In contrast, when the spatial information is combined with gradient information, the registration accuracy is improved, and the difference between a floating and reference image is minimal.

The proposed registration method based on HOG shows good computational efficiency. All experiments were performed in MATLAB 2019b. The hardware was a computer with the following technical performance: Inter (R) Core (TM) i7-8550U CPU @ 1.80 GHz; Memory (RAM) 8 GB DDR4; GeForce MX150 4 GB video; hard disk 500 GB SSD. The values of the computation time were provided under the same computing power for each registration stage and method (Table 1). Compared with the MI-based intensity method, the computation time for the proposed MI-based HOG increases by 6.22% for S1, 6.15% for S2, and 4.78% for S3, respectively. Moreover, Table 1 displays the quantitative MI results, as follows: the statistics before registration, after translation, and the statistics for global registration. The experimental results show that the registration based on the MI–HOG method improves the registration accuracy, while the computational efficiency remains almost unchanged.

There are some limitations with this study. The placement of Harris corners/fiducials can be sensitive to error because of the patient’s motion. Harris corner/fiducial sites are limited to those locations that correspond to anatomic landmarks of the brain. Moreover, our patient numbers were low. The registration accuracy may be influenced by the brain injuries like brain metastases, neurodegeneration, and stroke, and future work devoted to these items is planned.

## 4. Conclusions

The effect of the implementation details on the behavior of the similarity measure and its influence on the quality of registration was reported. A rigid and multi-modal registration method using a similarity measure based on image histogram-oriented gradients was proposed, in which both global changes of intensity information and spatial information are considered. Both sparse features refer to corners detected using the Harris corner detector operator and dense features as the image histogram of oriented gradients were used for image registration. Consistent with expectation, an improved registration precision and computational efficiency were obtained through a robust and user-independent similarity metric. To summarize, the registration based on MI–HOG similarity measure improved the registration accuracy by relying on both intensity stationarity from pixel to pixel and on gradient spatial information about the pixels. Another advantage of the proposed method is its simplicity in terms of computational complexity.

## Figures and Tables

**Figure 1 entropy-22-01299-f001:**
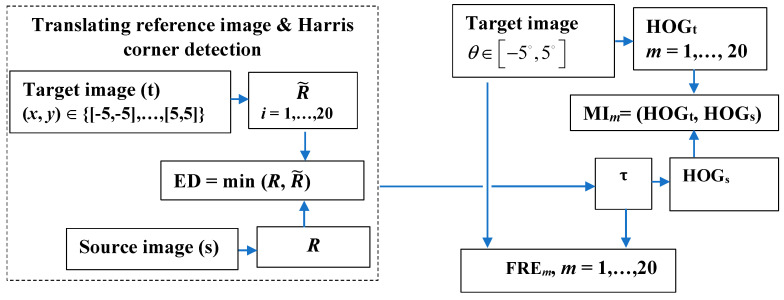
A flowchart of the proposed registration method. In the left block, the coarse registration step is displayed. The translation vector over the *x* and *y* axis; *R* and $\tilde{R}$ corner points in the source (s) and target/registered (t) images, respectively; and the Euclidian distance (ED) between similar strong corners are determined. *i* = 1, …, 20 is the incremental step for translational operation with a step of 0.5. In the right block, the fine registration based on the rotation is performed and histograms of gradients (HOGs, HOGt) for the source (s) and target (t) images are determined. Then, mutual information (MI) and fiducial registration error (FRE) are computed. *m* = 1, …, 20 denotes the rotation operations with a step of 0.5.

**Figure 2 entropy-22-01299-f002:**
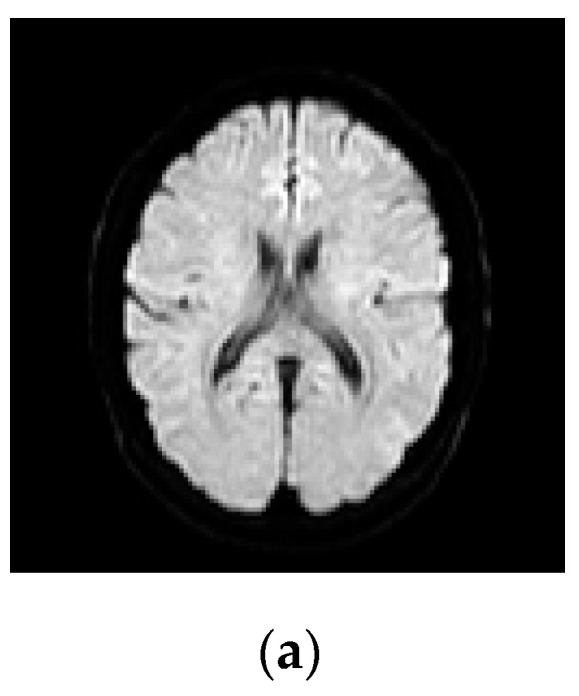
Examples of pairwise linear registration/translation between source and target images, for a healthy patient, using corner points detection. Image labeling using the Harris operator. Detected corners have similar locations and belong to the same regions, but show certain mobility. The optimal translation parameter for a coarse registration is identified as *τ* = (−4.5; −4.5). The fixed/source image is displayed in green and the moving/target image in magenta. (**a**) T2w. (**b**) DTI (*b* = 500 s/mm^2^). (**c**) T2w-DTI (*b* = 500 s/mm^2^). (**d**) DTI (*b* = 1250 s/mm^2^). (**e**) T2w-DTI (*b* = 1250 s/mm^2^).

**Figure 3 entropy-22-01299-f003:**
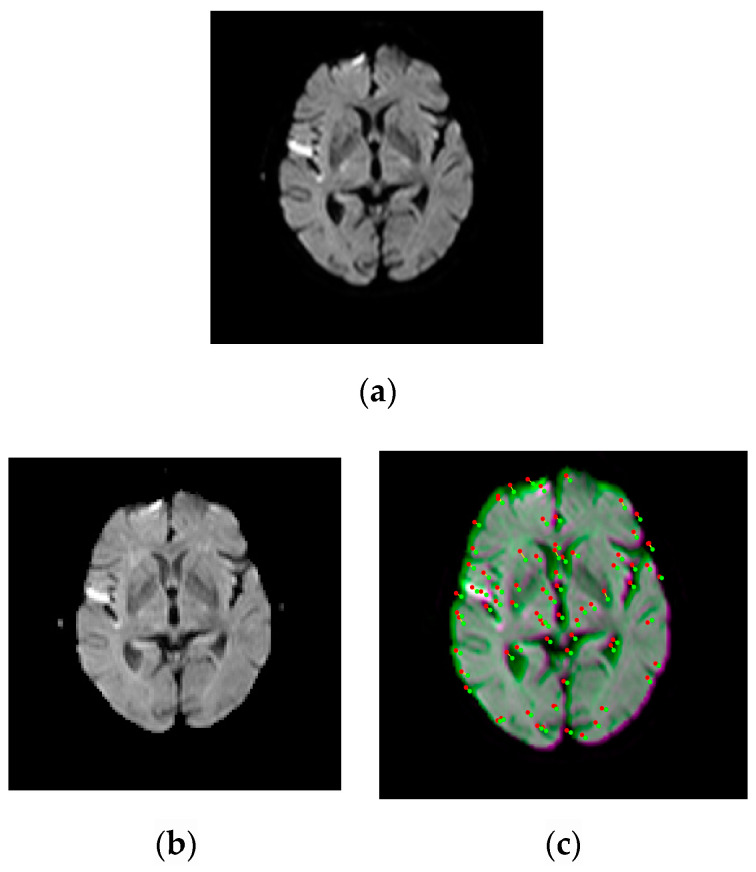
Examples of pairwise linear registration/translation between source and target images, for a patient with right parietal lobe hemorrhage, using corner points detection. Image labeling using the Harris operator. Detected corners have similar locations and belong to the same regions, but show certain mobility. The optimal translation parameter for a coarse registration is identified as *τ* = (−2.5; −2.5). The fixed/source image is displayed in green and the moving/target image in magenta. (**a**) T2w. (**b**) DTI (*b* = 500 s/mm^2^). (**c**) T2w-DTI (*b* = 500 s/mm^2^). (**d**) DTI (*b* = 1250 s/mm^2^). (**e**) T2w-DTI (*b* = 1250 s/mm^2^).

**Figure 4 entropy-22-01299-f004:**
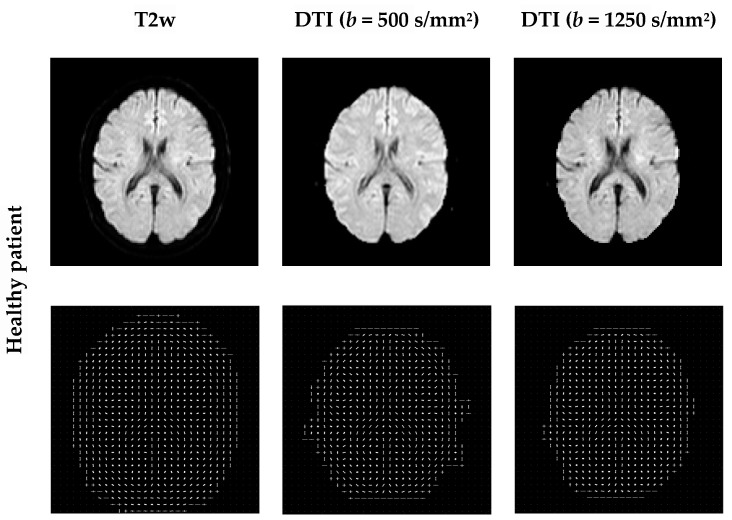
Examples of HOG features extracted from the original images. HOG computes block-wise histogram gradients with multiple orientations. An image is divided into 8 × 8 patches. A normalization of the gradients is performed by combining four 8 × 8 patches to create a 16 × 16 block for features extraction. A features vector was computed and 34,596 features are extracted.

**Figure 5 entropy-22-01299-f005:**
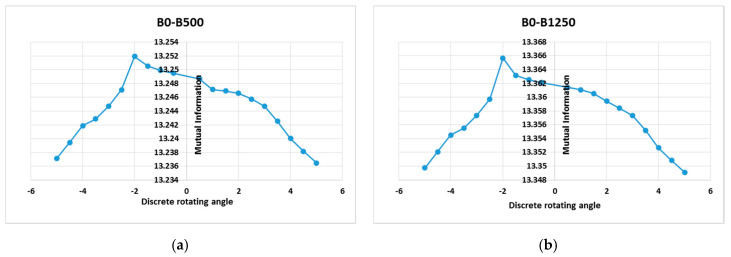
First row: MI based on histogram of gradients pattern for the healthy subject (S1), which depends on a rotation parameter in the range *θ*∈[−5°, +5°] with a step of variation of 0.5° and for a translational parameter of −4.5 pixels along the *x* and *y* axis. (**a**) T2w-DTI (*b*-values = 500 s/mm^2^) registration; (**b**) T2w-DTI (*b*-values = 1250 s/mm^2^) registration. These curves show a maximum at the correct points of alignment, i.e., (−2°; 13.2507) for the left plot and (−2°; 13.3658) for the right plot. Second row: MI based on histogram of gradients pattern for the diseased subject S2, which also depends on a rotation parameter in the range *θ*∈[−5°, +5°] with a step of variation of 0.5° and for a translational parameter of −2.5 pixel along the *x* and *y* axis. (**c**) T2-w-DTI (*b*-values = 500 s/mm^2^) registration; (**d**) T2-w-DTI (*b*-values = 1250 s/mm^2^) registration. These curves show a maximum at the correct points of alignment, i.e., (2°; 13.3501) for the left plot and (2°; 15.3682) for the right plot.

**Figure 6 entropy-22-01299-f006:**
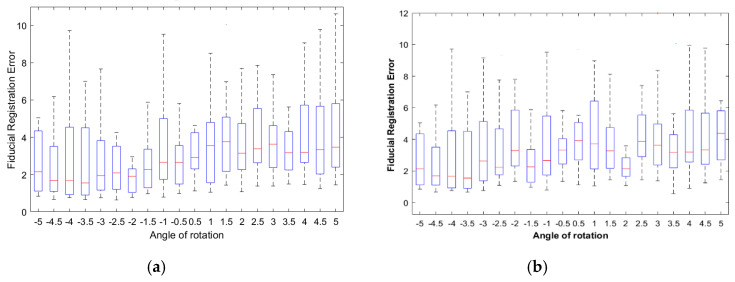
Box and whisker plot of the FRE between selected Harris corners as fiducials for T2w-DTI image registration. The registration accuracy in terms of FRE is performed for 30 control points/fiducials. (**a**) Subject S1; (**b**) subject S2; (**c**) subject S3.

**Figure 7 entropy-22-01299-f007:**
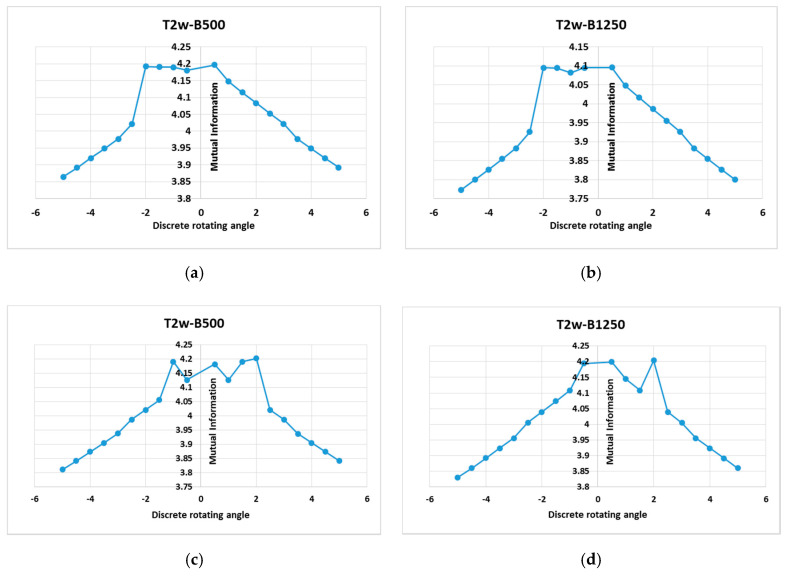
First row: MI pattern (as a statistical intensity relationship between corresponding pixels in source and target images) for the healthy subject S1, which depends on a rotation parameter in the range *θ*∈[−5°, +5°] with a step of variation of 0.5° and for a translational parameter of −4.5 pixels along the *x* and *y* axis. (**a**) T2w-DTI (*b*-values = 500 s/mm^2^) registration; (**b**) T2w-DTI (*b*-values = 1250 s/mm^2^) registration. These curves show two local maxima, i.e., (−2°; 4.781) and (+0.5°; 4.948) for the left plot, and (−2°; 4.734) and (+0.5°; 4.901) for the right plot. Second row: MI pattern (as a statistical intensity relationship between corresponding pixels in source and target images) for the diseased subject S2, which also depends on a rotation parameter in the range *θ*∈[−5°, +5°] with a step of variation of 0.5° and for a translational parameter of −4.5 pixel along the *x* and *y* axis. (**a**) T2w-DTI (*b*-values = 500 s/mm^2^) registration; (**b**) T2w-DTI (*b*-values = 1250 s/mm^2^) registration. These curves show a certain number of local maxima. As example, (+1.5°; 4.1902) and (2°; 4.2015) for the left plot, and (+0.5°; 4.1995) and (2°; 4.2041) for the right plot.

**Table 1 entropy-22-01299-t001:** Effect of brain image multi-registration on mutual information (MI) (mean ± SD) and computation time. HOG, histogram of oriented gradients.

	Registration Method		MI	Computation Time (s)
**Healthy patient**	Before registration		3.077 ± 0.163	2.33
	After translation/linear registration		4.638 ± 0.421	2.48
	After global rigid registration	MI-based intensities	4.018 ± 0.095	2.41
		MI-based HOG	13.357 ± 0.004	2.56
**Hemorrhagic patient**	Before registration		3.167 ± 0.172	1.95
	After translation/linear registration		4.728 ± 0.544	1.78
	After global rigid registration	MI-based intensities	4.133 ± 0.132	1.95
		MI-based HOG	15.355 ± 0.004	2.07
**Ischemic stroke**	Before registration		2.996 ± 0.235	1.85
	After translation/linear registration		5.120 ± 0.505	1.62
	After global rigid registration	MI-based intensities	4.247 ± 0.162	1.88
		MI-based HOG	14.145 ± 0.004	1.97

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
