# Peer review of "Combining Sparse and Dense Features to Improve Multi-Modal Registration for Brain DTI Images"

_entropy, 2020, doi:10.3390/e22111299_

Round 1
Reviewer 1 Report
Dear Authors,
I found the paper Combining sparse and dense features to improve multi-modal registration for brain DTI images interesting to read. However, there are some critical issues you should address:
- The description of the second part of the method (image rotation) is not precise enough. In Fig. 1, you use variables that you have not explained in the text.
- Lines 264-267: Mutual information is a measure that combines joint entropy of two variables and marginal entropies. None of the entropies involved (joint entropy and marginal entropies) itself can tell you how much information one variable has about the other. The sentence is very misleading.
- Lack of comments to figures. It will help if you guide the reader through the important information presented in figures. If there is no such thing in a figure, you can omit it.
- The most critical is experimental work. I am afraid you are not able to give proper conclusions when using only one patient. I advise you to widen the experimental work considerably.
Minor things:
- Please unify acronyms all over the paper. For example, the acronym T2w also appears as T2-w, T2-weighted, T2W.
- 3: Probabilities are commonly designated with non-capital letters.
- The designation of corner points begins with the letter H, the same as entropy, which brings some unnecessary ambiguity.
- After Eq. 4, you state that \Omega=30, in Fig. 2, the number is 20.
- 1: explain the meaning of HOG_s and HOG_t and \tau(x).
- 2: what is the meaning of green and purple color. Explain to the reader what to look for in figure.
- 3, Fig. 4: comment on the figures in the main text.
- Line 272: less à smallest?
- It is not clear what you mean by traditional MI. Please define it and make a clear distinction to the proposed method.
- You should avoid using apostrophes. When you put some words in apostrophes like “gold standard” in line 288, the reader cannot interpret them.
Best Regards, Reviewer
Author Response
I would like to express thanks to the reviewers for their valuable and constructive comments concerning our submission.
The manuscript has been revised to address the issues raised in the review process. There were added supplementary information on the Harris corners, data from a second patient and some references were changed. Accordingly, the content of certain figures has been improved. Furthermore, it was thoroughly edited to eliminate language issues and enhance readability wherever warranted.
Now I am going to reply to review comments one by one as listed below. According to the comments, we have revised the manuscript and the revised parts are highlighted as blue color in the text.

Reviewer 2 Report
The authors present a method for multi-modal registration to register T2-W and DTI images. The combine sparse and dense features to get better registration results.
The major area for improvement is the inclusion of prior work in the broad area of DTI registration. Including these papers in the current paper both in the prior art and also in comparing the results of the proposed method with the prior art which is already published would make the paper more compelling and more complete and comprehensive. In my quick search on this topic I could find more than three papers that have already demonstrated similar registration tasks of DTI registration. (See attached Prior Art) - for example - , Alena Uus (Biomedical Image Registration. 2020 ), Roura E ( Funct Neurol. 2015;)
The second major area for improvement is the lack of inclusion of Deep Learning based methods. It would be very interesting to compare the performance of the proposed method against Deep Learning based DTI Registration, which again has been published already - for example - , Alena Uus (Biomedical Image Registration. 2020 ), Roura E ( Funct Neurol. 2015;). (Please see attached prior art).
The third major area for improvement is the inclusion of established registration frameworks to compute baseline and demonstrate the results of the proposed method against those baselines.
A few suggestions for established registration frameworks are
1. MRtrix3 (MRtrix3: https://www.mrtrix.org) toolbox.
2. Elastix (https://elastix.lumc.nl/)
3. AIRLAb (https://github.com/airlab-unibas/airlab)
4. Anatomical Normalization Tools (ANTs) (http://stnava.github.io/ANTs/)
The fourth major scope for improvement is that the current work demonstrates the registration on one healthy patient. The authors include this in the limitations of the study. It has to be noted that researchers have already attempted to register T1w and DTI in presence of atrophy in the context of multiple sclerosis Roura E ( Funct Neurol. 2015;). Hence extending the work to include some disease context would make the work more compelling.
The fifth major comment is the sequential approach to first estimate the translation parameter and then the rotation parameter. Most solutions that I have come across estimate rigid transformation (joint translation and rotation) - for example Elastix estimates rigid transformation. It is not sufficiently motivated why the translation is first estimated and then the rotation parameter is then determined. It is also not clear and suffitiently motivated as to the small search space (range) of translation and rotation parameters which have been considered in this study(Translation 0 to 5 pixel and Rotation of [-5° to 5°] degrees.
Is it because the images can get transformed in that specific range only ? If so why ?
The sixth major comment is the realization of the registration seems (from the way it is described and presented in the paper) to be like a grid search and/or brute force search and not an optimization based approach. This could be because the search space is limited and also not minimized (solved) jointly
The seventh major comment is that there could be utilization (or alteast comparison) of more recent feature detectors which have been proven to work well for medical image registration. For example MIND descriptors (https://www.sciencedirect.com/science/article/abs/pii/S1361841512000643?via%3Dihub)
The remaining comments below could also be addressed to make the paper more compelling.
- The references are sometimes not chronologically presented
a. Reference 10 is first presented which is published 2019 and then Reference 11 is cited which is published in 2017
b. Reference 19 is published in year 2019 and then Reference 20 is cited which is published in year 2018 - Some of the key references for registration which are elaborated in the prior art are very old. Reference 13 and 14 are from year 2013
- In the contributions the author claim -
"The translational and rotational parameters of the rigid transformation followed by HOG features extraction are iteratively generated by the MI in order to find best matches between reference registered images."
In the methods section the translation is independently computed without HOG or MI
There seems to be some inconsistencies and could be better articulated - The selection of fiducial points could be better explained. How is it guaranteed that the same points would get detected in both the images is not clear ?
How are the fiducials evenly spread over the whole brain is not clear. - The difference between fiducials for FRE error computation and sparse features for registration is not clear. Do these points sets overlap or are they disjoint sets of points ?
- The number of images used is also not well captured. In Dataset section the authors state 15 pairs of T2w and DTI. What are the (transformation) changes in these 15 datasets is not captured. Is it over some time period? Then in Results section it is stated three stacks of 15 images. This leads to some inconsistency in understanding
- It could be included how many features were extracted or detected in Section 3, for estimating the translation.
- Some studies could be performed on synthetically transforming the images with known values of translation and rotation, to have more comprehensive results and also comparing it against known ground truth values
- Figure 2 has only translation motion, While Figure 3 has Rotation motion.
It is unclear what would happen when there is large rotation motion and when the method tries to estimate the translation parameter only - Figure 4 : It is not clear how HOG features are extracted in the background region as well?
- As claimed in abstract if the proposed method has similar accuracy then what is the main motivation for this work ?
- "similar registration accuracy and robustness with the classical registration algorithm" - - Some sentences could be reworded to improve the English
- faces the challenge of a higher order variation of anatomical features into the image stack.
- these are possible with the cost of slices misalignment due to motion
- MR image modalities like T2-w and DTI brain images to be analyzed in the same
coordinate system, especially to correct the motion such as, breathing, bladder filling or cardiac motion (Brain images would not have breathing or cardiac motion) - Some Typos could be corrected
- The acquisition matrix was 128 128.
- The parameters of transformation are the stablished translation

Author Response
I would like to express thanks to the reviewers for their valuable and constructive comments concerning our submission.
The manuscript has been revised to address the issues raised in the review process. There were added supplementary information on the Harris corners, data from a second patient and some references were changed. Accordingly, the content of certain figures has been improved. Furthermore, it was thoroughly edited to eliminate language issues and enhance readability wherever warranted.
The recommended papers are cited and the results are comparatively studied. However, it is not possible to carry out a new set of experiments at this stage since time is of the essence. The recommended experiments will be conducted and its results will be reported in a future issue of the journal. We are thankful to the Reviewer very much for the valuable suggestions.
Now I am going to reply to review comments one by one as listed below. According to the comments, we have revised the manuscript and the revised parts are highlighted as blue color in the text.

Round 2
Reviewer 1 Report
Dear Authors,
You have addressed many issues. However, you can still improve a lot. Firstly, you lack consistency in variable designation (text, equations), missing description of variables, etc. Secondly, writing needs additional proofing. Thirdly, the most critical remains experimental work. With the analysis of an additional person, you only slightly improved the report. To show the proposed method's wide applicability and its robustness, you should analyze ten and more subjects.
Best Regards, Reviewer
Author Response
We are very grateful to reviewer 1 for their comments. Our responses are uploaded in a pdf file. Thank you

Reviewer 2 Report
The authors have incorporated many of the recommendations suggested by the reviewer.
I still think that comparing the results of the proposed method with the prior art which is already published would make the paper more compelling and more complete and comprehensive. This is especially for the recent work that use Deep Learning for DTI registration for example - Irina Grigorescu et al (WBIR 2020), Alena Uus (Biomedical Image Registration.2020 )
Would it be possible to get at least one baseline using any of the existing registration tools (especially that use Deep Learning based methods) (in short time) ?
As suggested earlier few suggestions for established registration frameworks are
1. MRtrix3 (MRtrix3: https://www.mrtrix.org) toolbox.
2. Elastix (https://elastix.lumc.nl/)
3. AIRLAb (https://github.com/airlab-unibas/airlab)
4. Anatomical Normalization Tools (ANTs) (http://stnava.github.io/ANTs/)
Author Response
We are very grateful to reviewer 2 for their comments. Our responses are uploaded as a pdf file. Thank you.
